# Effects of Dietary n-3 Polyunsaturated Fatty Acids and Selenomethionine on Meat Quality and Fatty Acid Composition in Finishing Pigs

**DOI:** 10.3390/foods14071124

**Published:** 2025-03-24

**Authors:** Yunju Yin, Hu Zhang, Teng Hui, Ran Li, Hong Chen, Minquan Xia, Bin Feng, Yong Yang, Yaowen Liu, Zhengfeng Fang

**Affiliations:** 1Laboratory for Animal Disease Resistance Nutrition of the Ministry of Education, Animal Nutrition Institute, Sichuan Agricultural University, Chengdu 611134, China; yinyj1124@163.com (Y.Y.); zhu68842@gmail.com (H.Z.); fengbin@sicau.edu.cn (B.F.); 2Laboratory for Food Science and Human Health, College of Food Science, Sichuan Agricultural University, Ya’an 625014, China; huiteng@sicau.edu.cn (T.H.); liran@sicau.edu.cn (R.L.); chenhong945@sicau.edu.cn (H.C.); 13339732276@163.com (M.X.); yangyong676@163.com (Y.Y.); yaowenliu@sicau.edu.cn (Y.L.); 3Laboratory of Agricultural Product Processing and Nutrition Health (Co-Construction by Ministry and Province), Ministry of Agriculture and Rural Affairs, Ya’an 625014, China

**Keywords:** n-3 polyunsaturated fatty acids, SeMet, fatty acids, meat quality, antioxidant

## Abstract

The interaction between selenomethionine (SeMet) and n-3 polyunsaturated fatty acids (n-3 PUFA) in producing n-3 PUFA-enriched pork remains unknown. This study investigates the effect of different n-3 PUFA sources (linseed oil vs. fish oil) and SeMet supplementation on meat quality and fatty acid composition in finishing pigs. Key findings demonstrate that dietary supplementation with 0.3 mg/kg SeMet significantly enhances the *L**_24h_ value (lightness) of the longissimus thoracis et lumborum (LTL) tissue compared to 3% linseed oil or fish oil treatments alone (*p* < 0.05). Pork flavor improvement is further supported by increased serine content (*p* < 0.05) and a notable tendency toward elevated total sweet amino acids (Thr + Ser + Gly + Ala + Pro) in LTL tissue (*p* = 0.077). Compared with 3% sunflower oil (control group), 3% linseed oil or fish oil significantly enhances n-3 PUFA content while reducing the n-6/n-3 ratio in both LTL and subcutaneous adipose tissue (*p* < 0.05). The synergistic interaction between SeMet and oil (linseed oil or fish oil) is observed, increasing α-linolenic acid (ALA; C18:3n-3), eicosatrienoic acid (C20:3n-3), and total n-3 PUFA deposition in subcutaneous fat tissue (*p* < 0.05). SeMet increases the activities of total superoxide dismutase (T-SOD) and catalase (CAT). Meanwhile, the SeMet-fish oil combination decreases lipids oxidation compared to individual treatments (*p* < 0.05). Collectively, 3% linseed oil or fish oil effectively enhances unsaturated fatty acid profiles, while concurrent SeMet addition may synergistically enhance certain nutritional attributes (improved oxidative stability) and sensory scores (enhanced *L*_24 h_* value and flavor precursors). We, therefore, recommend adding 0.3 mg/kg SeMet to the n-3 PUFA-enriched pork production process.

## 1. Introduction

Cardiovascular diseases (CVDs) have emerged as the predominant public health challenge in China, with incidence and mortality rates showing a concerning upward trajectory alongside rapid economic development and dietary pattern transitions. Current epidemiological data reveal a staggering CVDs mortality rate of one death every 10.5 s, known as the “number one killer” of human health [1,2]. This epidemic is strongly associated with modern nutritional imbalances, particularly the combination of excessive saturated fat intake and insufficient consumption of omega-3 polyunsaturated fatty acids (n-3 PUFA)—essential nutrients demonstrated to modulate cholesterol metabolism and attenuate CVDs progression [3]. One way to increase n-3 PUFA intake without changing consumers’ nutritional behavior is to fortify traditional foods such as meat and meat products with n-3 PUFA.

Emerging evidences highlighted the feasibility of n-3 PUFA enrichment in pork through dietary interventions. Supplementation with 3% linseed oil in swine diet increased the n-3 PUFA content while it decreased the n-6/n-3 ratio from 15 to 2.5 in muscle tissue [4]. However, dosage optimization is critical, as excessive linseed oil (e.g., 5.0%) produces undesirable flavor and reduces oxidative stability [5]. Studies on fish oil supplementation have demonstrated dose-dependent effects: while 0.5% supplementation effectively increased long-chain n-3 PUFA (n-3 LC-PUFA), escalating to 8% quadrupled n-3 LC-PUFA content but the meat was softer [6,7]. As Zaloga [8] emphasized, elevated n-3 PUFA deposition heightened susceptibility to lipid oxidation, accelerating rancidity and reduced both sensory quality and product shelf-life.

Selenium (Se), an essential trace element, functions as a critical component of hepatic glutathione peroxidase (GSH-Px) and selenoproteins, mitigating lipid oxidation by scavenging free radicals through GSH-Px-mediated antioxidant pathways [9]. The biological efficacy of Se supplementation is critically dependent on its chemical form, with organic sources such as selenomethionine (SeMet) exhibiting enhanced bioavailability and tissues deposition compared to inorganic forms (e.g., sodium selenite, SeNa) [10]. A study demonstrated that dietary 0.3 mg/kg SeMet supplementation in finishing pigs elevated tissue Se deposition, improved oxidative stability, and enhanced meat quality [11]. Furthermore, Se-enriched (0.25 mg Se/kg) diets of finishing pigs reduced volatile basic nitrogen (TVB-N) and microbial spoilage, effects attributed to its antioxidative properties [12]. However, it remains unclear whether Se can effectively counteract lipid peroxidation induced by n-3 PUFA and how their combined supplementation affects meat quality. Therefore, this study investigates the interactive effects of n-3 PUFA sources (linseed oil and fish oil) and SeMet on growth performance and meat quality in finishing pigs, providing theoretical and practical insights for producing n-3 PUFA-enriched pork.

## 2. Materials and Methods

### 2.1. Animals and Experiments Design

The linseed oil was bought from Defu Oil Co., LTD.(Ya’an, China), and the ALA content was 54%. The fish oil contained 21% DPA + DHA, and was bought from Yikang Natural Flavor Oil Refinery (Ji’an, China). SeMet was provided by Sichuan Xinyimei Biotechnology Co., Ltd. (Mianyang, China); the product is named Selenide ™2000. The main ingredients are L-SeMet and carrier thinner (zeolite powder, medical stone, silica, and rice husk powder). The L-SeMet content is ≥5000 mg/kg, and the actual Se content is ≥2000 mg/kg.

A total of 60 finishing pigs (Duroc × Landrace × Large, female) with an average weight of 73.00 ± 2.99 kg were divided into 5 groups with 12 animals per group, distributed in four finishing housings. To mitigate the environmental influences, each treatment was systematically distributed across three spatial zones (anterior, central, posterior) within the finishing housing, ensuring equally scattered in all housing units. The diets for the 5 groups were as follows: (a) 3% sunflower oil supplement (CON); (b) 3% linseed oil supplement (LO); (c) 3% fish oil supplement (FO); (d) 3% linseed oil supplement with 0.3 mg/kg of SeMet (LSe); (e) 3% fish oil supplement with 0.3 mg/kg of SeMet (FSe) (Table 1).

The CON group received 3% sunflower oil, selected for its extremely low levels of n-3 PUFA while maintaining comparable saturated (SFA), monounsaturated (MUFA), and polyunsaturated (PUFA) fatty acid ratios to linseed oil. The sunflower oil in the CON group was substituted with 3% linseed oil and fish oil. SeMet products with an actual Se content of 0.3 mg/kg were added to the LSe and FSe groups. To counteract potential lipid peroxidation from increased oils intake, all diets added 200 IU/kg vitamin E as antioxidant.

The basal diet was formulated according to the NRC [13], and the diets at the same phase were isoenergetic, isonitrogen, and isofat in all experimental groups. The formulation of the experiment diets and measured values of fatty acid composition are shown in Table 2 and Table 3. The feeding experiment was divided into two phases according to body weight (Phase 1, 75–100 kg, and Phase 2, 100–135 kg stage), and lasted 52 days.

### 2.2. Sample Collection

At the end of the feeding experiment, 6 pigs in each group were selected based on the principle of similar average weight. The selected pigs were slaughtered after a 24 h fasting period. A total of 10 mL of blood from the anterior venous of each pig was collected, left for more than 30 min at 4 °C, centrifuged for 15 min at 3500 r/min, and the upper serum was taken and divided into EP tubes and stored at −20 °C for testing. Muscle samples were taken immediately after slaughter and skin removal. The longissimus thoracis et lumborums (LTL) of the left half of the carcass from the 5th–6th segments of the lumbar spine was taken; cut two muscle blocks of about 50 g from top to bottom, perpendicular to the muscle fibers, and stored at −20 °C for testing. Samples of the LTL and subcutaneous fat were collected, divided into frozen storage tubes, wrapped in tin foil, and stored at −80 °C for testing (Figure 1).

### 2.3. Growth Performance Measurement

Daily feed intake was recorded during feeding, and all pigs were weighed on 0, 26, and 52 d of the experiment to calculate the average daily feed intake (ADFI), body weight gain (BWG), average daily gain (ADG), and the ratio of feed intake to body weight gain (F/G).

### 2.4. Serum Biochemical Indexes

Triglyceride (TG), total cholesterol (TC), low-density lipoprotein cholesterol (LDL-C), and high-density lipoprotein cholesterol (HDL-C) in serum were determined by a fully automatic biochemical analyzer (HITACHI 3100, Tokyo, Japan) according to the instructions of kits (Nanjing Jiancheng Bioengineering Institute, Nanjing, China) at the Institute of Animal Nutrition, Sichuan Agricultural University.

### 2.5. Meat Quality Measurement

Dissected LTL tissue was used to determine meat quality, considering meat color, pH, marbling score, drip loss, and cooking loss. At 45 min, 24 h, 48 h, and 7d postmortem, meat color and pH were measured. Meat color was measured using the CIE Lab color space system (*L**: lightness; *a**: redness; *b**: yellowness), using a calibrated colorimeter 3nh chromameter (NR60CP, 3nh Co., Ltd., Guangzhou, China) with the following settings: CIE 10° standard observer, D65 illuminant, and Φ8 mm aperture. Before measurement, the instrument was calibrated using a standard whiteboard. Each sample was measured at three different locations and the average was taken as the final result. pH values were determined using a food pH meter (Testo 205, Testo, Schwarzwald, Germany). Each sample was measured at three different locations and the average value was taken [14]. After the loin-eye blocks were stored at 0–4 °C for 24 h, the marbling scores were subjectively evaluated according to the marbling score map (marbling from 1 to 10, with 1 = no marbling and 10 = overly abundant marbling), and the average of values from different observers was used for each muscle sample.

The LTL samples (100 g) were stripped of the fat and fascia, weighed before cooking, then randomly allocated to a thermostatic water bath at 80 °C and cooked in the same batch to an internal temperature of 70 °C. After cooling to room temperature and weighing again, the percentage of cooking loss was then calculated using the following formula:Cooking loss (%) = (Initial weight − Final weight)/Initial weight (g) × 100%(1)

For drip loss determination, muscle samples were cut into 5 × 3 × 2.5 cm^3^ cuboids along the muscle fibers and weighed at 45 min postmortem, suspended in bags at 4 °C for 24 h, 48 h, and 7 d, and reweighed. The percentage of drip loss was then calculated using the following formula:Drip loss (%) = (Initial weight − Final weight)/Initial weight (g) × 100%(2)

### 2.6. Muscle Chemical Composition

The moisture, crude protein (CP), and intramuscular fat (IMF) content in the LTL tissue was detected by a previous study [15]. Briefly, about 50 g muscle samples were weighed and placed into a Labconco freeze dryer (model 2.5, Labconco Corp., Kansas, MO, USA) with a temperature of −45 °C and a vacuum of less than 10 μm of Hg. Samples were freeze-dried for 48 h, and then the muscle samples were reweighed to calculate the content of moisture. Dried muscle samples were subsequently pulverized using a hammer mill, then the Kjeldahl nitrogen determination was used to determine the CP content, and the cable extraction method (SOX416, Sox-therm, North Rhine-Westphalia, Germany) was used to determine the IMF content.

### 2.7. Antioxidant Capacity Measurement

The catalase (CAT), total superoxide dismutase (T-SOD) activity, and the malondialdehyde (MDA) content in serum and muscle were determined according to the instructions of the kit (Nanjing Institute of Biological Engineering, Nanjing, China).

### 2.8. Amino Acid

The amino acid in the LTL tissue was determined by an automatic amino acid analyzer, as described previously [16] with minor modifications. The freeze-dried meat was about 150 mg in a hydrolysis tube, and 15 mL of 6.0 mol/L hydrochloric acid was added. The tube was filled with nitrogen, sealed, and hydrolyzed in a 110 °C oven for 24 h. Then, it was cooled and transferred to a 50 mL volumetric bottle, filled with ultra-pure water, and shaken well. Then, we absorbed 200 μL into the amino acid vial and freeze dried it under negative pressure before adding 800 μL of 0.02 mol/L hydrochloric acid for resolution. The 0.22 μm microporous filter membrane was filtered and analyzed with an automatic amino acid analyzer (Hitachi L-8900, Tokyo, Japan). The amino acids were expressed as % of dry matter.

### 2.9. Fatty Acid Composition Measurement

Lipids in the LTL and subcutaneous fat tissue samples were extracted as previously described [17] with a minor modification [18]. Total lipids in the upper layer were collected and determined by a GC 2010 plus model gas chromatogram. Chromatographic conditions: HP-88 (100 m × 0.25 mm × 0.20 μm) was used. The sample volume was 1.0 μL, and the inlet temperature was 240 °C. The detector was FID, and the temperature was 280 °C. We programmed the temperature rise: it was kept at 100 °C for 13 min, then heated up to 180 °C at 10 °C/min for 6 min, then heated up to 192 °C at 1 °C/min for 9 min, then heated up to 230 °C at 3 °C/min for 10 min. The carrier gas was nitrogen, and the flow rate was 1.3 mL/min. The fatty acids in the LTL tissue were expressed as mg/100 g fresh weight, while subcutaneous fat tissue was expressed as g/100 g fresh weight.

### 2.10. Data Statistics and Analysis

The experimental data were processed using Excel 2021 for preliminary organization, with statistical analyses conducted in SPSS 29.0 (IBM, New York, NY, USA). Normality was verified through Shapiro–Wilk tests, and homogeneity of variance was confirmed via Levene test. The one-way ANOVA was used to analyze the groups supplemented with different oils (CON, LO, and FO). The source of n-3 PUFA and whether SeMet was added (LO, FO, LSe, and FSe) were analyzed by two-factor ANOVA with a general liner model (GLM), and the significance of the main effect and interaction effect were analyzed. If the difference was significant, the least significant difference (LSD) was used for multiple comparison. The results were expressed as mean, and the standard error (SEM) of the mean was used to represent the variability of the values. *p* < 0.05 was a significant difference, *p* ≤ 0.01 was an extremely significant difference, and 0.05 ≤ *p* ≤ 0.1 was a trend.

## 3. Results

### 3.1. Dietary Fatty Acid Composition

Table 3 shows the fatty acid composition of the experimental diets. Compared to the CON diet, both LO and FO diets exhibited elevated total n-3 PUFA content and reduced n-6/n-3 ratio, with LO showing significantly higher n-3 PUFA content than FO. The n-3 PUFA in the LO diet was dominated by α-linolenic acid (ALA; C18:3n-3), whereas FO diets contained predominantly eicosapentaenoic acid (EPA; C20:5n-3) and docosahexaenoic acid (DHA; C22:6n-3). The SFA/MUFA/PUFA ratio in the LO and CON diets were comparable, with low EPA and DHA contents. Conversely, the FO diet demonstrated elevated SFA and MUFA ratios compared to both the LO and CON diets.

### 3.2. Growth Performance

The growth performance of finishing pigs is listed in Table 4. During the 100–135 kg phase, pigs fed the FO diet exhibited a significantly elevated feed-to-gain ratio (F/G) compared to both the CON and LO groups (*p* < 0.01). Similarly, over the 75–135 kg phase, the F/G of the FO group remained elevated compared to the CON group (*p* < 0.05). A marginal reduction in body weight gain (BWG) was observed in the FO group during the 100–135 kg phase (*p* = 0.098). Main effect analysis further indicated that FO supplementation markedly increased F/G during the 100–135 kg phase (*p* < 0.01) and showed a tendency toward higher F/G compared to the LO group over the 75–135 kg phase (*p* = 0.06). In contrast, dietary SeMet supplementation had no significant effect on BWG, average daily gain (ADG), average daily feed intake (ADFI), or F/G at any growth phase (*p* > 0.05).

### 3.3. Serum Biochemical Indexes

We examined the effects of n-3 PUFA and SeMet on serum biochemical indexes of finishing pigs. Figure 2A illustrates that during the 75–100 kg phase, the LO and FO groups exhibited reduced HDL-C level, while the LO group showed lower TC level than the CON group (*p* < 0.05). Main effect analysis indicated that SeMet increased the serum TC level (*p* < 0.05) and showed a tendency to increase HDL-C (*p* = 0.057) and LDL-C (*p* = 0.054) levels. During the 100–135 kg phase, the FO group increased serum LDL-C level compared with the LO group (*p* < 0.05), and tended to increase TC level (*p* = 0.082) (Figure 2B). n-3 PUFA and SeMet exhibited an interactive effect on serum LDL-C and TC levels (*p* < 0.01). The FSe group displayed higher LDL-C level than the LO, FO, and LSe groups, while maintaining lower TC level than the FO and LSe groups (Figure 2C).

### 3.4. Meat Quality

The meat quality and muscle chemical composition (IMF, CP, and moisture) are shown in Figure 3 and Table 5. Two-way ANOVA revealed no significant interactive effects between dietary treatments (oil or SeMet) and postmortem time (45 min, 24 h, 48 h, and 7 d) on meat quality indexes (*p* > 0.05) (Figure 2). Neither SeMet nor n-3 PUFA supplementation significantly influenced muscle chemical composition or meat quality indexes (*p* > 0.05). However, a significant interaction on n-3 PUFA and SeMet was identified for 24 h *L** value, where the LSe group demonstrated higher *L** values compared to the LO and FSe groups (*p* > 0.05) (Figure 3D).

### 3.5. Antioxidant Capacity

We tested the antioxidant capacity of the LTL tissue and serum. The serum antioxidant capacity is displayed in Figure 4. During the 75–100 kg phase, both FO and LO supplementation decreased serum T-SOD activity compared to the CON group, with FO increasing serum MDA content compared to the CON and LO groups. Main effect analysis showed that FO decreased T-SOD activity and increased MDA content compared to the LO supplementation, while SeMet increased the CAT activity (*p* < 0.05). Two-way ANOVA revealed that the interaction of FO and SeMet increased MDA content (*p* < 0.05). During the 100–135 kg phase, LO and FO reduced the T-SOD and CAT activities compared to the CON group (*p* < 0.01). FO and SeMet co-supplementation increased MDA content and T-SOD activity (*p* < 0.05).

In the LTL tissue, the T-SOD activity was higher in the FO group than that in the CON group (*p* < 0.05), and the main effect analysis indicated that SeMet decreased T-SOD activity (*p* < 0.01). However, no significant differences were observed in MDA content and CAT activity among all groups (*p* > 0.05).

### 3.6. Amino Acids Composition

We examined the amino acid composition of the LTL tissue of finishing pigs. As shown in Table 6, SeMet tended to enhance sweet amino acids (Thr + Ser + Gly + Ala + Pro) (*p* = 0.077) and Ser content (*p* < 0.05). However, other amino acid levels showed no discernible change (*p* > 0.05).

### 3.7. Fatty Acid Composition

Fatty acid composition in the LTL and subcutaneous fat tissue was detected. As shown in Table 7, the erucic acid (C22:1n-9) content was higher in the FO group than that in the LO group (*p* < 0.05). LO-fed pigs demonstrated significantly higher ALA and eicosatrienoic acid (C20:3n-3) than the CON and FO groups (*p* < 0.01). Furthermore, in comparison with the CON and LO groups, FO supplementation increased the EPA and DHA contents (*p* < 0.01), while the LO group had higher EPA content than the CON group. Both LO and FO groups showed higher n-3 PUFA content and lower n-6/n-3 ratio than the CON group (*p* < 0.01). There were no significant differences observed in the SFA, MUFA, PUFA, and n-6PUFA contents among the CON, LO, and FO groups (*p* > 0.05). The two-way ANOVA analysis confirmed that the LO group showed greater levels of C20:3n-3, ALA, and PUFA/SFA (*p* < 0.05), whereas the FO group exhibited higher levels of EPA, DHA, heneicosanol (C21:0), eicosenoic acid (C20:1 n-9), and C22:1 n-9 (*p* < 0.05). Neither SeMet supplementation nor its interaction with n-3 PUFA significantly affected the fatty acid composition of the LTL tissue of finishing pigs.

Fatty acid composition in subcutaneous fat tissue is shown in Table 8. The ALA content was elevated in the LO and FO groups compared to the CON group, with LO demonstrating greater ALA than FO (*p* < 0.01). Both the LO and FO groups decreased linoleic acid (LA; C18:2n-6) and increased EPA content compared with the CON group (*p* < 0.01). Additionally, compared with the CON and FO groups, C20:3n-3 content increased in the LO group (*p* < 0.01). The DHA content was higher in the LO group than the CON and LO groups (*p* < 0.01). The LO group showed lower C20:3n-6 content than the CON and FO groups (*p* < 0.01). The SFA content was higher in the FO group than that in the CON and LO groups (*p* < 0.01), with MUFA content being increased in the FO group compared to the CON group (*p* < 0.05). Compared to the CON and LO groups, PUFA content decreased in the FO group (*p* < 0.01). The n-3 PUFA content increased in the LO and FO groups compared to the CON group, and it was noticeably more than twice as high in the LO group (*p* < 0.01). Both the LO and FO groups decreased the n-6 PUFA content and n-6/n-3 ratio compared with the CON group (*p* < 0.01), and the LO group had a lower n-6/n-3 ratio than the FO group (*p* < 0.01). The PUFA/SFA ratio was the highest in the LO group, and was lower in the FO group than that in the CON group (*p* < 0.01).

Two-way ANOVA revealed that when FO was used as the source of n-3 PUFA, the C14:0, C15:0, C17:0, C21:0, C23:0, C16:1, and C20:1n-9; C22:1n-9, EPA, DHA, SFA, and MUFA contents; and n-6/n-3 ratio were significantly elevated compared to the LO group (*p* < 0.05). Contrastingly, LO supplementation significantly enhanced the ALA, C20:3n-3, PUFA, n-3PUFA, and PUFA/SFA ratio. In addition, SeMet co-supplementation elevated adipose deposition of ALA, C20:3n-3, C16:0, C18:0, C18:1n-9c, C20:1n-9, C20:2n, SFA, MUFA, and n-3PUFA (*p* < 0.05). Notably, the interaction of SeMet and LO significantly increased the contents of ALA and gamma-linolenic acid (GLA, C18:3n-6) (*p* < 0.05) (Figure 5).

## 4. Discussion

n-3PUFA is abundant in unsaturated double bonds, particularly EPA and DHA, and its oxidation stability is suboptimal [19]. In terms of the primary fatty acid composition of feed, FO was predominantly composed of EPA and DHA, whereas LO primarily contained ALA. EPA and DHA had more unsaturated double bonds than ALA, and FO supplementation induced greater oxidative stress than LO, evidenced by the higher level of the lipid oxidation product MDA and lower T-SOD activity in the FO group. Dietary supplementation of 3% FO increased F/G during the 100–135 kg phase, which might be explained by the fact that FO easily impairs the intestinal antioxidant capacity of finishing pigs, which decreases nutrient utilization efficiency [20,21]. Interestingly, 0.3 mg/kg SeMet increased T-SOD activity without significantly affecting growth performance, which was consistent with previous findings [11], suggesting that SeMet partially eliminated the decrease in antioxidant damage caused by FO. Another study found that a dietary addition of 8% FO resulted in a decrease in ADG and an increase in backfat thickness of finishing pigs [7]. However, there was no statistically significant difference in BWG or ADG arising from the low supplemental level of FO and diets with identical fat content.

We also noticed a decrease in serum TG level in the LO group, which was in line with recent research [22]. EPA and DHA reduce serum TG level by activating the transcription factor peroxisome proliferator activating receptor (PPARα) and inhibiting sterol reaction element binding protein 1 (SREBP-1) [20]. However, we failed to find no significant effect of LO on TG level; this might be explained by the higher SFA level in FO, in addition to EPA and DHA, which mutually restricted the regulation of serum TG level [23]. In line with Czech et al. [24], both LO and FO decreased the serum HDL-C level. FO and SeMet co-supplementation elevated serum LDL-C and TC levels. In mammals, SFA can stimulate de novo synthesis of TC, which increases the serum TC level. Combined with the regularity of fatty acid deposition in tissues, serum TC and LDL-C levels increased as a result of higher SFA content in the FO diet [25].

The purchasing decisions of consumers are also influenced by traditional meat quality indexes when considering n-3 PUFA-enriched pork. Customers notice meat color directly through their senses, showing a preference for meat that is lighter and more red (higher *L** and *a** values) and less yellow (lower *b** value). The meat color stability was reduced due to the presence of double bonds in LO, which is prone to myoglobin and lipid oxidation during storage [26]. With the addition of SeMet, the *L** value in the LSe group was higher compared to the LO group, indicating that SeMet mitigated the adverse effect of LO on the *L** value of pork. When SeMet and LO were combined, SeMet functioned as an antioxidant to counteract the effect induced by LO, inhibit the cross-linking damage of lipid oxidation byproducts (such as malondialdehyde) to proteins, delay muscle tissue browning, and maintain brightness [27]. Other studies reported no effect of SeMet on meat color [27]. Juniper et al. [28] suggested that once the Se content of tissue exceeded the requirements of antioxidant enzymes, a further increase in tissue Se does not result in noticeable improvement in meat quality. In this study, SeMet had no effect on other meat color indexes, which may possibly be explained by the supplementation period and dose. In addition, theoretically, FO was expected to exhibit higher oxidative stress level compared to LO, a condition that could potentially compromise muscle tissue stability [29]. However, our findings revealed no significant alterations in meat color parameters, despite the observed differences in oxidative status. Meanwhile, muscle tissue analysis demonstrated higher T-SOD activity in the FO group. In contrast to the findings in serum, the main effect analysis unexpectedly showed that SeMet decreased T-SOD activity in muscle tissue. We postulate that this phenomenon may reflect an adaptive response, where prolonged exposure to lipid oxidation triggers systemic redistribution of T-SOD to muscle tissue as a compensatory antioxidant mechanism. The antioxidative effect of SeMet showed a U-shaped curve, with excess potentially promoting oxidative mechanisms [30]. This study showed that n-3 PUFA treatment had no significant difference in umami and bitter amino acids content in muscle tissue, whereas SeMet increased Ser content. This observation might be attributed to the fact that the methionine in SeMet regulated the amino acid metabolism of pigs, consequently influencing the amino acid composition of pork. However, amino acid composition of muscle remained unaffected by different source of n-3 PUFA. It was evident that SeMet had the potential to enhance the amino acid favorable for pork flavor [31].

Our investigation revealed distinct tissue-specific modulation of fatty acid profiles through n-3 PUFA supplementation, which selectively regulated n-3 PUFA composition in both LTL and subcutaneous adipose tissue, with differential effects between lipid sources. LO supplementation mainly increased the ALA and C20:3n-3 content, while FO predominantly increased the EPA and DHA content. These modifications significantly improved the lipid profile, with a reduced n-6/n-3 ratio and an increased PUFA/SFA ratio. These changes could be explained by the competitive enrichment of n-3 PUFA inhibiting the carbon chain extender and desaturase required for n-6 PUFA synthesis [32]. Notably, we observed significant tissue-specific deposition efficiency, with adipose tissue showing 5.67 times higher n-3 PUFA deposition in the LO group compared to the CON group, 2.85 times higher observed in muscle. This differential deposition pattern aligns with the unique lipid absorb and metabolism of monogastric animals, where dietary PUFA directly deposited into tissue phospholipids and triglycerides the body without hydrogenation, indicating a strong correlation between the fatty acid composition of the diet and the fatty acid composition of the animal body [33]. The results of fatty acid analysis in various treatments were in agreement with the regularity of fatty acid deposition in various diets. Interestingly, despite comparable dietary C20:3n-3 content across groups, compared with the CON and FO groups, the LO group exhibited 2.9- and 5.7-times increases in muscle and adipose C20:3n-3 content, respectively, which might be explained by the fact that C20:3n-3 is a metabolic intermediate in ALA elongation/desaturation pathways and a direct precursor for EPA and DHA [34]; thus, C20:3n-3 content in tissues was correlated with ALA content in diets. Our findings demonstrated that while LO supplementation significantly elevated EPA and DHA content in muscle and adipose tissue, which could be because the experimental diet included more lipid supplements, ALA in sunflower oil or LO can still increase DHA content to some extent through fatty acid metabolism. DHA level showed no statistical differentiation between the LO and CON groups. This specific disparity persisted despite higher ALA in the LO diet compared to the sunflower oil-based CON diets. The findings demonstrated that while ALA conversion to long-chain n-3 PUFA in finishing pigs significantly enhanced EPA deposition, its capacity to improve DHA synthesis remained insignificant. This constrained DHA conversion efficiency is strongly correlated with dietary ALA level and the initial body weight. Notably, low-dose LO supplementation (<2%) marginally elevated porcine DHA synthesis [35], a phenomenon potentially attributable to ALA’s competitive inhibition of Δ6-desaturase activity against C24:5n-3 [36], the essential precursor for DHA biosynthesis [37].

As an animal-derived lipid source, FO inherently contains higher SFA and MUFA content. Compared with LO, FO supplementation showed higher SFA, MUFA content, and n-6/n-3 ratio, and lower PUFA, n-3PUFA content, and PUFA/SFA ratio in the adipose tissue, while only the PUFA/SFA ratio was lower in muscle tissue, potentially attributable to inefficient n-3PUFA deposition in the muscle. Additionally, it is worth noting that SeMet enhanced adipose deposition of ALA and C20:3n-3, indicating improved n-3 PUFA deposition capacity. This dual mechanism involves (1) antioxidant protection of SeMet redirecting fatty acids from oxidation towards storage, and (2) synergistic lipid metabolism regulation through combined SeMet-LO action [38,39]. Although SeMet potentially elevated DHA synthesis, the degree of unsaturation in FO exceeded the antioxidative capacity of SeMet at a level of 0.3 mg/kg, resulting in non-significant differences. Surprisingly, the efficacy of SeMet in promoting n-3 PUFA deposition was adipose-specific, likely due to the high fatty acid content of adipose tissue stimulating SeMet to regulate lipid metabolism and enhance its antioxidant capabilities [40].

Data from Chinese National Bureau of Statistics indicate an annual per capita pork consumption of 30.5 kg (≈83.56 g/d) [41]. Based on standard lean pork composition (55% muscle, 27% fat), this equates to approximately 67% muscle and 33% fat intake [42]. Referencing Chinese Nutrition Society guidelines [43], daily recommended intakes for ALA, EPA + DHA, and total n-3 PUFA are 1333.33, 250, and 1111.11 mg, respectively. Our analyses demonstrated that 3% LO-supplemented pork provided 156%, 24%, and 217% of these respective requirements. Given that the conversion efficiency of ALA in the body is about 5–15% [44], LO-derived pork delivers functionally adequate n-3 PUFA levels to meet recommendations. Comparatively, 3% FO-supplemented pork met 41%, 230% and 106% of daily requirements. While FO pork provided twice the EPA + DHA recommendation, its ALA content remains insufficient. Given that ALA is an essential fatty acid and metabolic precursor, pork produced with 3% LO showed superior nutritional value by meeting n-3 PUFA demand through both direct provision and conversion potential. Consequently, in terms of nutritional value, it surpasses that of pork produced with FO. However, during the production of n-3 PUFA-enriched pork, the supplementation of linseed oil and fish oil may inhibit fat deposition, resulting in lean carcass and reduced marbling, and the energy–protein ratio needs to be adjusted to maintain the growth rate [45]. However, pork enriched with high levels of n-3 PUFA remains more susceptible to lipid oxidation during storage, necessitating the optimization of packaging conditions. This functional meat product holds significant appeal for health-conscious consumers, particularly those at high risk of cardiovascular disease. Clear and Codex-compliant nutritional labeling (including EPA + DHA content labeling), along with consumer education initiatives emphasizing the importance of optimizing the n-6/n-3 ratio, may enhance consumer purchase intent.

## 5. Conclusions

Dietary n-3 PUFA supplementation can reduce antioxidant capacity without adversely affecting meat quality, and 0.3 mg/kg SeMet co-supplementation further enhances antioxidant status. There is a positive correlation between the diet and the fatty acid composition of pork. Linseed-oil-derived ALA increases the EPA content while failing to significantly elevate DHA. Therefore, increasing the DHA content in pork necessitates exogenous fortification with preformed DHA sources such as fish oil. Nonetheless, for optimal health benefits, linseed oil exhibits relative superiority, and it is advisable to consider incorporating sufficient antioxidant additives such as SeMet.

## Figures and Tables

**Figure 1 foods-14-01124-f001:**
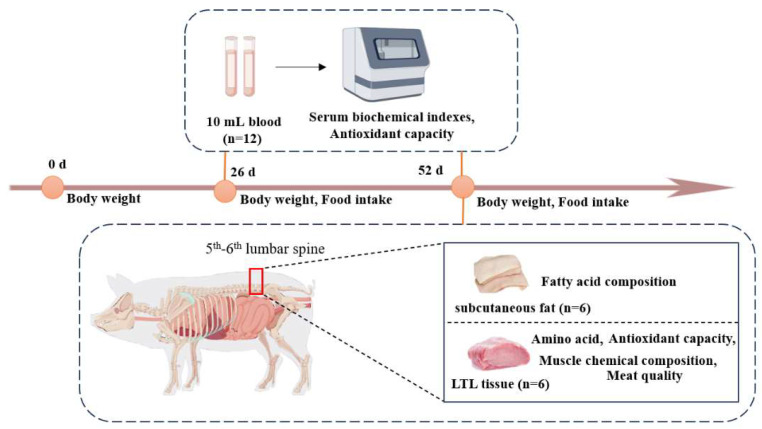
Sample collection and index determination.

**Figure 2 foods-14-01124-f002:**
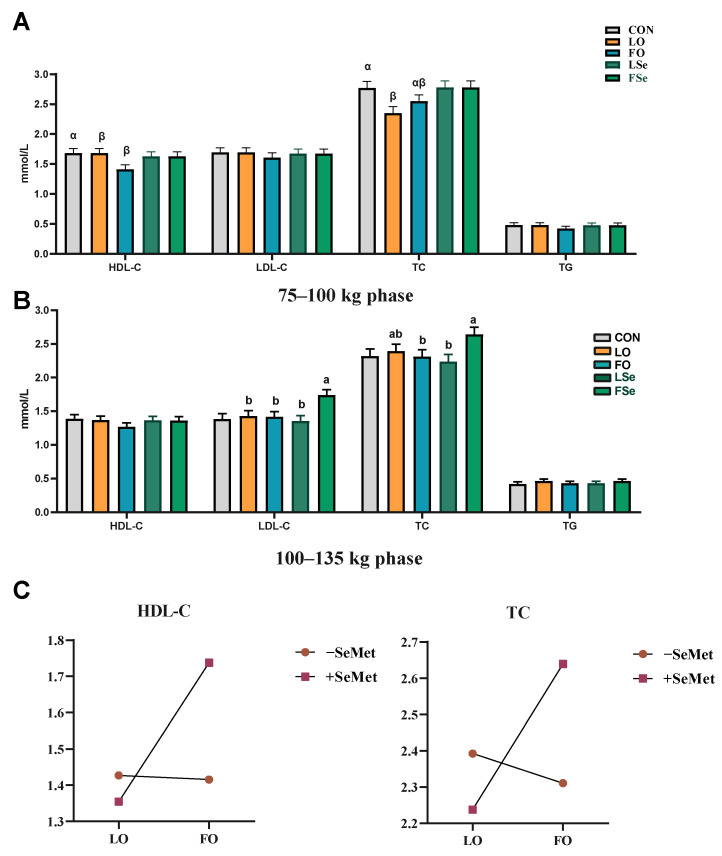
Effect of n-3 PUFA and SeMet addition on serum biochemical indexes of finishing pigs. (**A**) 75–100 kg phase, (**B**) 100–135 kg phase, and (**C**) interaction of oil and SeMet. Note: TG: triglyceride; TC: total cholesterol; LDL-C: low-density lipoprotein cholesterol; HDL-C: high-density lipoprotein cholesterol; CON: control group; LO: linseed oil; FO: fish oil; LSe: linseed oil + SeMet; FSe: fish oil+ SeMet. Data are expressed as means ± SEM. ^α,β^ Bars without a common superscript indicate significant difference (*p* < 0.05). When there is an interaction effect (*p* < 0.05), multiple comparisons are made between LO, FO, LSe, and FSe; the results are marked as a and b, and the different labels in the peer group indicate significant difference.

**Figure 3 foods-14-01124-f003:**
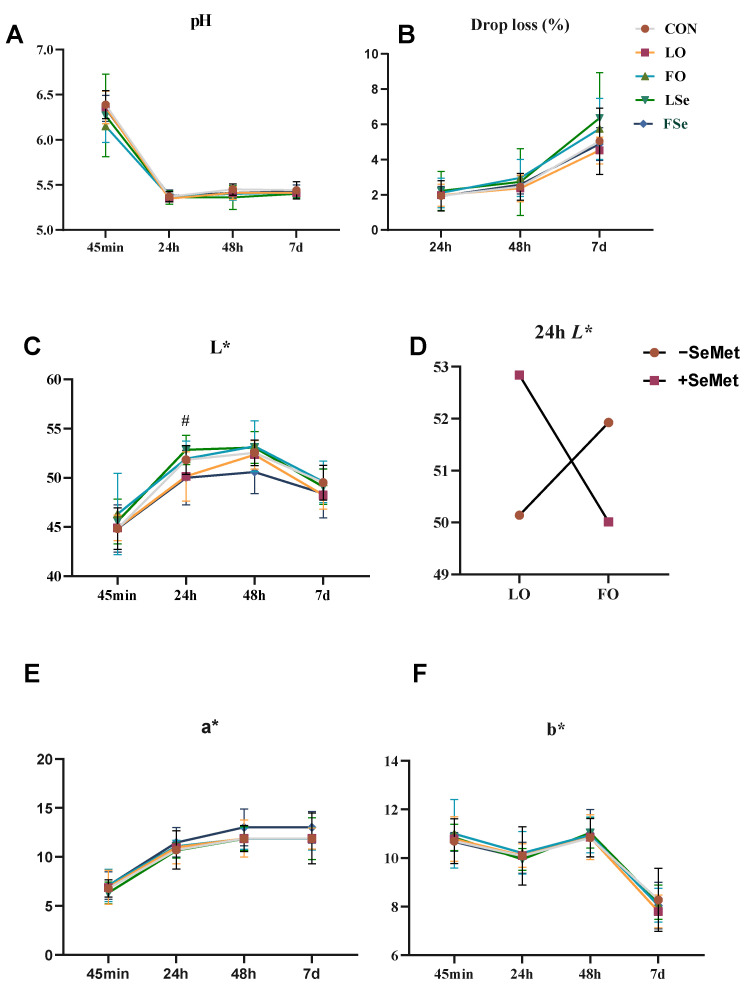
Effect of n-3PUFA and SeMet addition on meat quality of finishing pigs. (**A**) pH, (**B**) drip loss, (**C**) lightness (*L**), (**D**) interaction of oil and SeMet, (**E**) redness (*a**), and (**F**) yellowness (*b**). Note: CON: control group; LO: linseed oil, FO: fish oil, LSe: linseed oil + SeMet, FSe: fish oil + SeMet. Data are expressed as means ± SEM. # indicate significant difference (*p* < 0.05).

**Figure 4 foods-14-01124-f004:**
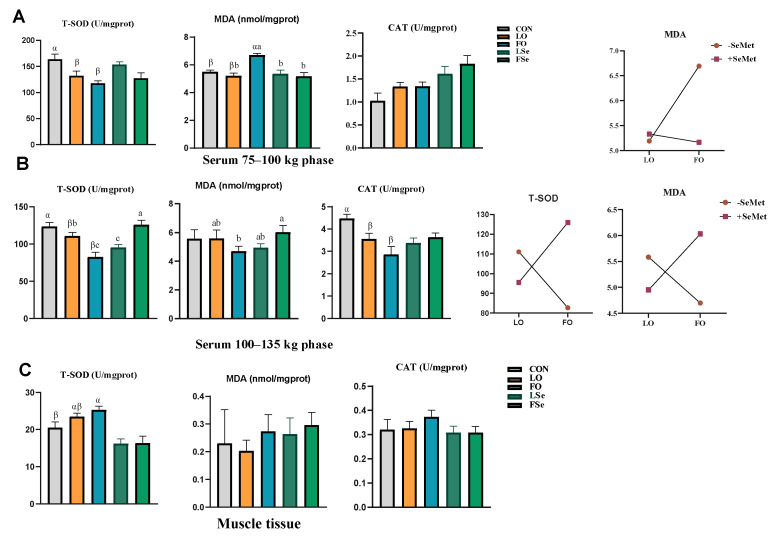
Effect of n-3PUFA source and Se addition on antioxidant indexes in finishing pigs (**A**) 75–100 kg phase, (**B**) 100–135 kg phase, and (**C**) muscle tissue. Note: CON: control group, LO: linseed oil, FO: fish oil, LSe: linseed oil + SeMet, FSe: fish oil + SeMet. Data are expressed as means ± SEM. ^α,β^ Bars without a common superscript indicate significant difference (*p* < 0.05). When there is an interaction effect (*p* < 0.05), multiple comparisons are made between LO, FO, LSe, and FSe; the results are marked as a, b, and c, and the different labels in the peer group indicate significant difference.

**Figure 5 foods-14-01124-f005:**
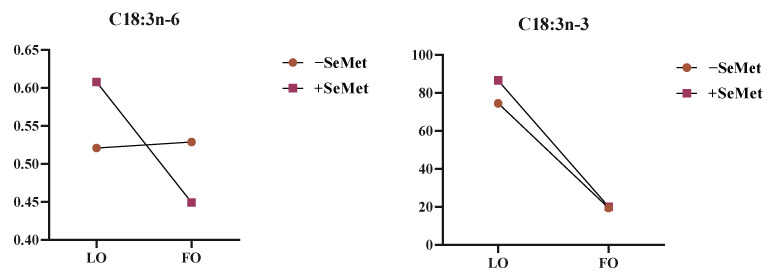
The interaction of different oils and SeMet on C18:3n-6 and C18:3n-3. Note: LO: linseed oil, FO: fish oil.

**Table 1 foods-14-01124-t001:** Experimental design.

Treatment	Diet Design
CON	3% sunflower oil diet
LO	3% linseed oil diet
FO	3% fish oil diet
LSe	3% linseed oil + 0.3 mg/kg SeMet
FSe	3% fish oil + 0.3 mg/kg SeMet

**Table 2 foods-14-01124-t002:** Composition and nutrient levels of basal diets (% of dry matter).

	Phase	75–100 kg	100–135 kg
Item			
Corn		72.00	75.12
Soybean meal		10.38	7.00
Wheat bran		7.16	7.64
Linseed cake meal		5.00	5.00
Sunflower oil		3.00	3.00
CaHPO_4_		0.85	0.74
CaCO_3_		0.65	0.61
Sodium chloride		0.30	0.30
Choline chloride (50%)		0.10	0.10
L-Lys·HCl (98.5%)		0.36	0.31
DL-Met (99%)		0.01	0.00
L-Thr (98.5%)		0.09	0.08
L-Trp (98%)		0.02	0.02
Mineral premix ^1^		0.03	0.03
Vitamin premix ^2^		0.03	0.03
Vitamin E ^3^		0.02	0.02
Total		100.00	100.00
Nutrients levels ^4^ (%)			
Digestion energy DE, Mcal/kg		3.40	3.41
Crude protein CP, %		13.15	11.96
SID Lys, %		0.73	0.62
SID Met, %		0.21	0.19
SID Thr, %		0.47	0.42
SID Trp, %		0.14	0.12
Ca, %		0.52	0.47
Standard Total Tract Digestible Phosphorus STTD-P, %		0.24	0.22

Note: ^1^ The mineral premix provides the following per kilogram of diet: Fe (FeSO_4_·H_2_O), 40 mg; Cu (CuSO_4_·5H_2_O), 3 mg; Zn (ZnSO_4_·H_2_O),50 mg; Mn (MnSO_4_·H_2_O), 2 mg; I (KI), 0.14 mg; Se (Na_2_SeO_3_), 0.15 mg. ^2^ The vitamin premix provides the following per kilogram of diet: VA 9000 IU; VD_3_ 3000 IU; VE 24 IU; VK_3_ 3 mg; VB_1_ 3 mg; VB_2_ 7.5 mg; VB_6_ 3.6 mg; VB_12_ 0.36 mg; pantothenate,15 mg; folic acid,1.5 mg; biotin, 1.5 mg. ^3^ Vitamin E is added in the form of DL-alpha-tocopherol acetate and the conversion relationship is 1 mg DL-alpha-tocopherol acetate = 1 IU VE. ^4^ Nutritional levels are calculated except for crude protein.

**Table 3 foods-14-01124-t003:** Dietary major fatty acid composition % (% of total fatty acids).

	Treatment	75–100 kg			100–135 kg		
Fatty Acid		CON	LO	FO	CON	LO	FO
C12:0		0.01	0.01	0.04	0.01	0.01	0.04
C14:0		0.14	0.14	2.46	0.13	0.19	2.33
C16:0		10.22	9.79	15.04	10.52	10.72	15.66
C17:0		0.10	0.12	0.39	0.10	0.12	0.38
C18:0		3.75	3.96	3.75	3.60	4.42	4.25
C20:0		0.26	0.22	0.36	0.25	0.22	0.35
C16:1		0.11	0.15	3.53	0.15	0.26	3.31
C17:1		0.05	0.05	0.11	0.04	0.05	0.11
C18:1n-9t		0.01	0.01	0.04	0.01	0.02	0.04
C18:1n-9c		24.98	21.29	21.49	25.10	23.19	22.37
C20:1n-9		0.02	0.01	0.10	0.02	0.01	0.09
C22:1n-9		0.03	0.27	2.32	0.02	0.11	1.92
C18:2n-6		55.15	32.98	30.51	54.59	32.45	30.07
C18:3n-6		0.02	0.11	0.12	0.02	0.10	0.12
C20:2n-6		0.03	0.05	0.27	0.05	0.06	0.26
C20:3n-6		0.00	0.02	0.08	0.00	0.03	0.08
C18:3n-3		4.08	29.92	6.13	4.40	26.87	6.19
C20:3n-3		0.01	0.02	0.02	0.01	0.08	0.17
C20:5n-3		0.30	0.26	5.28	0.32	0.29	4.73
C22:6n-3		0.01	0.07	7.03	0.01	0.08	6.33
SFA ^1^		15.07	14.75	22.82	15.20	16.38	24.00
MUFA ^2^		25.23	21.82	27.69	25.39	23.66	28.00
PUFA ^3^		59.70	63.43	49.49	59.41	59.96	48.00
n-3PUFA ^4^		4.40	30.27	18.45	4.75	27.32	17.43
n-6PUFA ^5^		55.2	33.16	30.98	54.66	32.64	30.53
n-6/n-3		12.55	1.10	1.68	11.51	1.19	1.75
PUFA/SFA		3.96	4.30	2.17	3.91	3.66	2.00
S/M/P ^6^		1:1.7:4	1:1.5:4.3	1:1.2:2.2	1:1.7:3.9	1:1.4:3.7	1:1.2:2

Note: All contents are measured values. CON: control group, LO: linseed oil, FO: fish oil, LSe: linseed oil + SeMet, FO + SeMet: fish oil + SeMet. ^1^ The sum of C4:0, C6:0, C8:0, C10:0, C11:0, C12:0, C13:0, C14:0, C15:0, C16:0, C17:0, C18:0, C20:0, C21:0, C22:0, C23:0, C24:0. ^2^ The sum of C14:1, C15:1, C16:1, C17:1, C18:1n-9t, C18:1n-9c, C20:1n-9, C22:1n-9, C24:1n-9. ^3^ The sum of C18:2n-6t, C18:2n-6, C18:3n-6, C18:3n-3, C20:2, C20:3n-6, C20:3n-3, C22:2, C20:5n-3, C22:6n-3. ^4^ The sum of C18:3n-3, C20:3n-3, C20:5n-3, C22:6n-3. ^5^ The sum of C18:2n-6t, C18:2n-6, C18:3n-6, C20:3n-6. ^6^ Refers to SFA: MUFA: PUFA, and the ratio is calculated with SFA as 1.

**Table 4 foods-14-01124-t004:** Effect of n-3 PUFA and SeMet addition on growth performance of finishing pigs.

Item	CON	LO	FO	LSe	FSe	SEM ^1^		*p*-Value			
						1	2	OTP ^2^	OIL	Se	OIL × Se ^3^
IBW	72.96	72.96	73.00	72.96	72.96	0.53	0.40	1.00	0.98	0.98	0.98
75–100 kg											
BW	104.42	104.63	103.96	103.38	103.71	0.84	0.67	0.95	0.91	0.59	0.72
BWG	31.95	31.55	31.82	30.42	32.15	0.41	0.38	0.92	0.20	0.60	0.34
ADG	1.23	1.21	1.22	1.17	1.24	0.02	0.02	0.93	0.19	0.61	0.34
ADFI	3.02	2.93	3.01	2.97	3.08	0.03	0.03	0.46	0.09	0.34	0.80
F/G	2.47	2.37	2.49	2.54	2.49	0.03	0.02	0.13	0.38	0.06	0.06
100–135 kg											
BW	132.42	134.00	131.63	132.67	132.08	1.36	1.11	0.78	0.52	0.85	0.70
BWG	29.94	30.63	28.59	29.86	30.15	0.54	0.41	0.10	0.30	0.63	0.17
ADG	1.15	1.18	1.10	1.15	1.16	0.02	0.02	0.11	0.32	0.60	0.17
ADFI	3.56	3.68	3.64	3.60	3.73	0.05	0.05	0.70	0.65	0.95	0.37
F/G	3.02_β_	3.10_β_	3.31_α_	3.11	3.16	0.04	0.03	<0.01	0.01	0.13	0.10
75–135 kg											
BWG	62.17	62.14	60.90	60.32	62.30	0.81	0.65	0.77	0.78	0.88	0.24
ADG	1.20	1.19	1.17	1.16	1.20	0.02	0.01	0.76	0.81	0.86	0.26
ADFI	3.20	3.17	3.26	3.33	3.39	0.05	0.04	0.73	0.32	0.07	0.87
F/G	2.73_β_	2.77_αβ_	2.87_α_	2.84	2.89	0.03	0.02	0.04	0.06	0.33	0.48

Note: Values are average values; *n* = 12. CON: control group, LO: linseed oil, FO: fish oil, LSe: linseed oil + SeMet, FO + SeMet: fish oil + SeMet, IBW: initial body weight, BW: body weight, BWG: body weight gain, ADG: average daily weight gain, F/G: feed to gain ratio. ^1^ 1 of SEM represents the SEM of CON vs. LO vs. FO of different oil groups, and 2 represents the SEM of 2 × 2 factor design with different n-3 PUFA source and SeMet. ^2^ OTP: Oil type, *p*-values from one-way ANOVA represent only different oil sources; multiple comparison results are marked with α and βat the lower right corner, with significant differences between different labels in the same line. ^3^ When there is an interaction effect (*p* < 0.05), multiple comparisons are made between LO, FO, LSe, and FSe.

**Table 5 foods-14-01124-t005:** Effect of n-3 PUFA and SeMet addition on the meat quality and muscle chemical composition longissimus thoracis et lumborums of finishing pigs.

Item	CON	LO	FO	LSe	FSe	SEM ^1^		*p*-Value			
						1	2	OTP ^2^	OIL	Se	OIL × Se ^3^
Cook loss, %	34.60	34.96	31.29	33.70	34.91	1.17	0.92	0.37	0.51	0.53	0.20
Marbling scores	3.58	3.17	3.42	2.83	3.42	0.29	0.20	0.84	0.38	0.72	0.72
Intramuscular fat, %	2.13	1.82	1.96	1.84	1.98	0.19	0.15	0.82	0.67	0.95	1.00
Crude protein, %	23.46	23.69	23.66	23.71	23.47	0.17	0.17	0.85	0.71	0.82	0.78
Moisture, %	72.33	72.05	72.30	71.78	72.65	0.23	0.23	0.88	0.24	0.93	0.50

Note: Values are average values; *n* = 6. CON: control group, LO: linseed oil, FO: fish oil, LSe: linseed oil + SeMet, FO + SeMet: fish oil + SeMet. ^1^ 1 of SEM represents the SEM of CON vs. LO vs. FO of different oil groups, and 2 represents the SEM of 2 × 2 factor design with different n-3 PUFA source and SeMet. ^2^ OTP: Oil type. ^3^ When there is an interaction effect (*p* < 0.05), multiple comparisons are made between LO, FO, LSe, and FSe.

**Table 6 foods-14-01124-t006:** Effect of n-3 PUFA source and SeMet addition on amino acids in the longissimus thoracis et lumborums of finishing pigs (% of dry matter).

Item	CON	LO	FO	LSe	FSe	SEM ^1^		*p*-Value			
						1	2	OTP ^2^	OIL	Se	OIL × Se ^3^
Asp	7.82	7.89	7.98	8.26	8.05	0.09	0.07	0.83	0.66	0.14	0.31
Thr	3.88	3.92	3.98	4.14	4.03	0.05	0.04	0.79	0.73	0.07	0.27
Ser	3.18	3.20	3.24	3.39	3.31	0.04	0.03	0.86	0.78	0.04	0.37
Glu	13.40	13.51	13.63	14.11	13.68	0.16	0.13	0.88	0.54	0.22	0.29
Gly	3.45	3.49	3.48	3.61	3.54	0.03	0.03	0.91	0.53	0.14	0.60
Ala	4.78	4.98	5.00	5.25	5.10	0.07	0.05	0.46	0.52	0.09	0.42
Val	4.22	4.26	4.32	4.49	4.36	0.05	0.04	0.78	0.64	0.13	0.27
Met	2.35	2.38	2.39	2.48	2.38	0.03	0.02	0.91	0.36	0.36	0.29
Ile	4.02	4.05	4.11	4.27	4.16	0.05	0.04	0.80	0.72	0.12	0.31
Leu	6.97	7.03	7.13	7.39	7.19	0.08	0.07	0.81	0.71	0.15	0.30
Tyr	3.63	3.68	3.74	3.86	3.81	0.04	0.04	0.60	0.94	0.10	0.43
Phe	3.38	3.44	3.48	3.60	3.54	0.04	0.03	0.68	0.82	0.10	0.45
Lys	7.51	7.57	7.66	7.97	7.75	0.09	0.07	0.84	0.67	0.10	0.29
His	3.70	3.75	3.78	3.89	3.93	0.05	0.04	0.41	0.73	0.12	0.97
Arg	5.62	5.69	5.79	6.02	5.84	0.07	0.06	0.72	0.71	0.08	0.22
Pro	3.14	3.19	3.32	3.37	3.38	0.05	0.04	0.39	0.43	0.19	0.48
Total ^4^											
Umami amino acids	21.22	21.40	21.61	22.38	21.72	0.24	0.20	0.87	0.58	0.19	0.29
Sweet amino acids	18.43	18.78	19.03	19.76	19.37	0.22	0.18	0.66	0.84	0.08	0.38
Bitter amino acids	37.77	38.19	38.65	40.11	39.15	0.41	0.36	0.78	0.73	0.11	0.33

Note: Values are average values; *n* = 6. CON: control group, LO: linseed oil, FO: fish oil, LSe: linseed oil + SeMet, FO + SeMet: fish oil + SeMet. ^1^ 1 of SEM represents the SEM of CON vs. LO vs. FO of different oil groups, and 2 represents the SEM of 2 × 2 factor design with different n-3 PUFA source and SeMet. ^2^ OTP: Oil type, *p*-values from one-way ANOVA represent only different oil sources. ^3^ When there is an interaction effect (*p* < 0.05), multiple comparisons are made between LO, FO, LSe, and FSe. ^4^ Umami amino acids = Asp + Glu; sweet amino acids = Thr + Ser+ Gly + Ala + Pro; bitter amino acids = Val + Ile + Leu + Phe + Lys + Arg + Met + His.

**Table 7 foods-14-01124-t007:** Effect of n-3 PUFA and SeMet addition on fatty acid composition of the longissimus thoracis et lumborums muscle of finishing pigs (mg/100g fresh weight).

Item	CON	LO	FO	LSe	FSe	SEM ^1^		*p*-Value			
						1	2	OTP ^2^	OIL	Se	OIL × Se^3^
SFA											
C12:0	1.10	0.79	1.14	1.06	0.93	0.20	0.12	0.58	0.69	0.91	0.36
C14:0	22.18	14.77	20.86	18.77	17.78	3.64	2.12	0.48	0.57	0.92	0.43
C16:0	439.03	309.26	475.53	430.81	393.06	68.94	49.52	0.44	0.54	0.85	0.33
C17:0	4.17	2.49	3.99	3.26	3.33	0.46	0.34	0.27	0.27	0.93	0.32
C18:0	254.05	152.78	261.06	230.49	208.79	36.69	28.20	0.35	0.46	0.83	0.28
C20:0	3.40	2.20	3.49	3.21	2.64	0.54	0.39	0.48	0.66	0.93	0.26
C21:0	1.27	0.47	1.76	0.59	1.16	0.22	0.21	0.11	0.03	0.56	0.37
C23:0	16.77	12.15	15.49	14.26	13.36	1.72	1.47	0.40	0.70	1.00	0.50
MUFA											
C16:1	41.55	38.25	57.70	50.71	46.79	11.13	5.88	0.35	0.53	0.95	0.35
C17:1	8.28	8.06	11.36	9.94	9.72	0.85	1.04	0.31	0.49	0.96	0.43
C18:1n-9t	0.96	0.71	0.96	0.92	0.85	0.18	0.11	0.67	0.71	0.82	0.51
C18:1n-9c	797.52	537.21	858.50	768.98	715.43	116.86	97.56	0.46	0.52	0.83	0.37
C20:1n-9	1.08	0.49	1.34	0.57	1.23	0.19	0.15	0.07	<0.01	0.97	0.73
C22:1n-9	0.21 _αβ_	0.15 _β_	0.43 _α_	0.17	0.36	0.04	0.04	0.02	<0.01	0.67	0.53
PUFA											
C20:2n	9.84	5.32	7.15	6.64	6.32	1.00	0.78	0.36	0.65	0.88	0.52
n-6PUFA											
C18:2n-6	245.58	164.88	203.86	211.17	170.74	21.28	19.28	0.49	0.99	0.87	0.34
C18:3n-6	1.15	0.95	1.35	1.14	1.02	0.13	0.11	0.30	0.53	0.76	0.24
C20:3n-6	4.16	3.28	5.17	4.08	4.37	0.50	0.44	0.21	0.23	1.00	0.37
n-3PUFA											
C18:3n-3	14.56 _β_	43.70 _α_	17.74 _β_	55.07	14.24	3.63	5.61	<0.01	<0.01	0.68	0.43
C20:3n-3	1.91 _β_	5.49 _α_	2.23 _β_	7.10	1.97	0.47	0.76	<0.01	<0.01	0.61	0.48
C20:5n-3	2.97 _γ_	12.04 _β_	34.50 _α_	14.79	29.96	3.01	3.04	<0.01	<0.01	0.86	0.47
C22:6n-3	3.39 _β_	3.71 _β_	28.30 _α_	4.26	23.68	2.60	2.97	<0.01	<0.01	0.61	0.52
Total											
SFA ^4^	750.98	504.31	792.30	708.21	645.86	111.79	82.34	0.41	0.52	0.87	0.32
MUFA ^5^	850.49	586.37	931.91	832.65	775.56	127.97	104.54	0.46	0.51	0.84	0.37
PUFA ^6^	283.71	239.76	300.56	304.51	252.53	24.70	28.33	0.70	0.94	0.89	0.35
n-3PUFA ^7^	22.82 _β_	64.94 _α_	82.77 _α_	81.21	69.85	7.24	7.92	<0.01	0.85	0.92	0.39
n-6PUFA ^8^	250.89	169.11	210.37	216.39	176.12	21.83	19.80	0.49	0.99	0.88	0.34
n-6/n-3	11.17 _α_	2.61 _β_	2.56 _β_	2.81	2.54	1.33	0.06	<0.01	0.22	0.47	0.40
PUFA/SFA	0.40	0.50	0.41	0.49	0.42	0.02	0.02	0.12	0.04	0.98	0.83

Note: Values are average values; *n* = 6. CON: control group, LO: linseed oil, FO: fish oil, LSe: linseed oil + SeMet, FO + SeMet: fish oil + SeMet. ^1^ 1 of SEM represents the SEM of CON vs. LO vs. FO of different oil groups, and 2 represents the SEM of 2 × 2 factor design with different n-3 PUFA source oils and SeMet. ^2^ OTP: Oil type, *p*-values from one-way ANOVA represent only different oil sources; multiple comparison results are marked with α, β, and γ at the lower right corner, with significant differences between different labels in the same line. ^3^ When there is an interaction effect (*p* < 0.05), multiple comparisons are made between LO, FO, LSe, and FSe. ^4^ The sum of C4:0, C6:0, C8:0, C10:0, C11:0, C12:0, C13:0, C14:0, C15:0, C16:0, C17:0, C18:0, C20:0, C21:0, C22:0, C23:0, C24:0. ^5^ The sum of C14:1, C15:1, C16:1, C17:1, C18:1n-9t, C18:1n-9c, C20:1n-9, C22:1n-9, C24:1n-9. ^6^ The sum of C18:2n-6t, C18:2n-6, C18:3n-6, C18:3n-3, C20:2, C20:3n-6, C20:3n-3, C22:2, C20:5n-3, C22:6n-3. ^7^ The sum of C18:3n-3, C20:3n-3, C20:5n-3, C22:6n-3. ^8^ The sum of C18:2n-6t, C18:2n-6, C18:3n-6, C20:3n-6.

**Table 8 foods-14-01124-t008:** Effect of n-3 PUFA and SeMet addition on fatty acid composition of subcutaneous fat in finishing pigs (mg/g fresh weight).

Item	CON	LO	FO	LSe	FSe	SEM ^1^		*p*-Value			
						1	2	OTP ^2^	OIL	Se	OIL × Se ^3^
SFA											
C12:0	0.44	0.43	0.50	0.51	0.50	0.03	0.02	0.24	0.45	0.25	0.26
C14:0	8.89 _β_	9.26 _β_	11.93 _α_	10.59	12.85	0.48	0.41	<0.01	<0.01	0.10	0.75
C15:0	0.28 _β_	0.23 _β_	0.49 _α_	0.27	0.47	0.02	0.03	<0.01	<0.01	0.65	0.25
C16:0	172.24 _β_	171.11 _β_	200.34 _α_	210.71	231.64	7.70	6.98	<0.01	0.04	<0.01	0.72
C17:0	3.64 _β_	2.92 _β_	6.75 _α_	3.57	5.92	0.37	0.39	<0.01	<0.01	0.83	0.10
C18:0	110.54	102.33	121.83	133.57	151.76	4.32	5.67	0.12	0.05	<0.01	0.94
C20:0	1.88	1.96	2.11	2.69	3.12	0.07	0.16	0.22	0.29	<0.01	0.60
C21:0	0.12	0.11	0.13	0.11	0.15	0.01	0.01	0.34	0.01	0.32	0.56
C23:0	1.02 _α_	0.64 _β_	0.97 _α_	0.65	0.85	0.06	0.04	<0.01	<0.01	0.33	0.20
MUFA											
C16:1	9.31 _β_	10.64 _β_	17.56 _α_	11.79	17.27	1.35	0.80	<0.01	<0.01	0.67	0.48
C17:1	1.16 _β_	1.02 _β_	1.60 _α_	1.20	1.44	0.08	0.06	<0.01	<0.01	0.90	0.07
C18:1n-9c	299.05 _β_	317.38 _αβ_	342.04 _α_	376.29	412.00	12.10	10.07	<0.01	0.05	<0.01	0.71
C20:1n-9	0.52 _β_	0.37 _γ_	0.75 _α_	0.49	1.26	0.04	0.09	<0.01	<0.01	<0.01	0.05
C22:1n-9	0.09 _β_	0.11 _β_	1.29 _α_	0.17	1.70	0.13	0.19	<0.01	<0.01	0.38	0.49
PUFA											
C20:2n	9.32 _α_	6.76 _β_	6.25 _β_	7.61	7.89	0.33	0.24	0.00	0.79	<0.01	0.35
n-6PUFA											
C18:2n-6	190.51 _α_	138.79 _β_	135.82 _β_	155.48	136.52	5.37	3.76	<0.01	0.14	0.24	0.28
C18:3n-6	0.52	0.52 ^b^	0.53 ^b^	0.61 ^a^	0.44 ^b^	0.02	0.02	0.97	0.06	0.92	0.04
C20:3n-6	0.80 _α_	0.57 _β_	0.88 _α_	0.60	0.91	0.04	0.04	<0.01	<0.01	0.45	0.91
n-3PUFA											
C18:3n-3	12.44 _γ_	74.46 _α_^b^	19.36 _β_^c^	86.56 ^a^	20.50 ^c^	5.61	6.49	<0.01	<0.01	0.02	0.05
C20:3n-3	1.71 _β_	9.80 _α_	2.14 _β_	11.91	2.76	0.75	0.93	<0.01	<0.01	0.03	0.21
C20:5n-3	0.38 _γ_	1.08 _β_	6.29 _α_	1.21	5.69	0.53	0.52	<0.01	<0.01	0.40	0.19
C22:6n-3	0.66 _β_	0.74 _β_	13.31 _α_	0.89	15.16	1.16	1.43	<0.01	<0.01	0.10	0.17
Total											
SFA ^4^	299.84 _β_	290.38 _β_	345.72 _α_	365.22	407.08	12.01	12.91	<0.01	0.03	<0.01	0.74
MUFA ^5^	310.35 _β_	329.88 _αβ_	363.93 _α_	390.27	433.97	13.41	10.65	<0.01	0.02	<0.01	0.76
PUFA ^6^	216.45 _α_	232.81 _α_	184.86 _β_	264.99	188.75	5.61	8.34	<0.01	<0.01	0.09	0.18
n-3PUFA ^7^	15.19 _γ_	86.07 _α_	41.10 _β_	100.58	43.66	6.33	5.62	<0.01	<0.01	<0.01	0.07
n-6PUFA ^8^	191.83 _α_	139.88 _β_	137.26 _β_	156.70	137.00	5.39	3.78	<0.01	<0.01	0.27	0.25
n-6/n-3	12.63 _α_	1.63 _γ_	3.34 _β_	1.56	3.15	1.38	0.18	<0.01	<0.01	0.24	0.56
PUFA/SFA	0.72 _β_	0.81 _α_	0.54 _γ_	0.74	0.47	0.03	0.03	<0.01	<0.01	0.06	0.93

Note: Values are average values; *n* = 6. CON: control group, LO: linseed oil, FO: fish oil, LSe: linseed oil + SeMet, FO + SeMet: fish oil + SeMet. ^1^ 1 represents the SEM of CON vs. LO vs. FO of different oil groups, and 2 represents the SEM of 2 × 2 factor design with different n-3PUFA source oils and SeMet. ^2^ OTP = Oil type, *p*-values from one-way ANOVA represent only different oil sources; multiple comparison results are marked with α, β, and γ at the lower right corner, with significant differences between different labels in the same line. ^3^ When there is an interaction effect (*p* < 0.05), multiple comparisons are made between LO, FO, LSe, and FSe. The results are marked in the upper right corner with a, b and, c, and the different labels in the peer group indicate significant differences. ^4^ The sum of C4:0, C6:0, C8:0, C10:0, C11:0, C12:0, C13:0, C14:0, C15:0, C16:0, C17:0, C18:0, C20:0, C21:0, C22:0, C23:0, C24:0. ^5^ The sum of C14:1, C15:1, C16:1, C17:1, C18:1n-9t, C18:1n-9c, C20:1n-9, C22:1n-9, C24:1n-9. ^6^ The sum of C18:2n-6t, C18:2n-6, C18:3n-6, C18:3n-3, C20:2, C20:3n-6, C20:3n-3, C22:2, C20:5n-3, C22:6n-3. ^7^ The sum of C18:3n-3, C20:3n-3, C20:5n-3, C22:6n-3. ^8^ The sum of C18:2n-6t, C18:2n-6, C18:3n-6, C20:3n-6.

## Data Availability

The original contributions presented in this study are included in the article. Further inquiries can be directed to the corresponding author.

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
