# Peer review of "Effects of Dietary n-3 Polyunsaturated Fatty Acids and Selenomethionine on Meat Quality and Fatty Acid Composition in Finishing Pigs"

_foods, 2025, doi:10.3390/foods14071124_

Round 1

Reviewer 1 Report

Comments and Suggestions for Authors

In the manuscript “Effects of Dietary n-3PUFA and SeMet on Meat Quality and Fatty Acid Composition of Finishing Pigs”, the authors analyzed the influence of pig feed rich in PUFA n-3 to improve the nutritional quality of meat, and the Selenium methionine antioxidant activity to prevent the lipid oxidation. The data collected are therefore important for estimating the intake of PUFA n-3 as in animal as in human, and for developing dietary enriched in PUFA; but the novelty, that means the utilization of SeMet as antioxidant and its interaction with the diets rich in PUFA have to explain better. Hence, the manuscript still needs some improvement.

  • It is suggested that the structure of the Introduction section should be adjusted appropriately, and to supplement some research progress directly related to the topic of this paper. In particular authors should be improve the part of SeMet.
  • Lines 6-8-9: remove “Key”
  • Line 17: insert the acronymous of longissimus thoracis et lumborums.
  • Line 19: insert the complete word of Ser.
  • Line 26: insert the complete word of
  • Line 26: for a better understanding of the sentence replace “MDA content” with “lipids oxidation”.
  • Line 52: insert the complete words of “DHA” and EPA”.
  • Line 62: remove the space.
  • Lines from 76 to 79: authors reported five group but four cages, why the finishing housings are not 5? Moreover authors reported 12 replicates per group and 1 pig per replicate, the sentence is confuse, it is suggest to replace "12 replicates" with "12 animals" and removed “and 1 pig per replicate“.
  • Lines from 90 to 93: these sentences are a repetition of the concept just stated. It is suggest removing it.
  • Line 97: remove “(2012).
  • Line 103: the % value is refer at dry matter or as is.
  • Lines 106-108 :remove the circle and add the space between (Na2SeO3) and 0.15 mg.
  • It is suggest a diagram to show all the sampling phases, to clarify how much, how and where the portions of meat were taken and used for the analyses.
  • Line 118: insert the acronymous for longissimus thoracis et lumborums.
  • Lines from 136 to 139: rewrite the sentences in clearer way.
  • Line 150: for a clearer reading it is recommended to insert the formula after analysis explanation.
  • Line 170: remove the colon.
  • Line 172: how authors have identified the peaks? And need to write the unit expression.
  • Line 182: how authors have identified the peaks? And need to write the unit expression.
  • Why authors do not report at statistical analysis for diet with storage time? in the tables are not easy to understand their interaction, it is suggest insert a separate table for diet * storage times.
  • Line 208: % of total fatty acids, authors means % of total FAME.
  • In table 3 is opportune to add the arachidonic acid.
  • Line 209: removed “LSe = linseed oil + SeMet; FSe =fish oil + SeMet”.
  • Lines 216, 236, 255, 289, 354, 372, 398, 517: enter the corresponding chapter numbers.
  • Lines 217 – 218: adjust the text format.
  • Line 218: insert the complete words of F/G.
  • Lines from 217 to 224: the sentences are confused, please rewrite in clear way.
  • In all tables are not clear where is the column for SeMet. It is suggested to change “Se” with SeMet. Moreover the column “Oil” is a repetition, with the same data of LO and FO present in column 2 and 3, so it is suggest to remove them.
  • Table 4: conform the text format in first row.
  • In “Serum biochemical indexes” result neither data have been reported for VLDL, if you have made the analysis, as reported in material and methods, the data for VLDL have to present, even if there is not significant effect, at least authors could insert in the text. Moreover in the chapter authors report "TC" but neither explanation about this parameter is report in material and methods.
  • Figure 1: insert the note of meaning of HDL-C, LDL-C, TC and TG.
  • Tables 4, 5, 6, 7, 8 and 9 need to be improves, they results unclear, and the table formatting is not consistent.
  • In table 5 in L* 24h is missing the significant mark.
  • In tables 7 and 8 we have to change “SAF” with the correct acronymous ”SFA”.
  • In table 7 and 8 it is suggest to insert the arachidonic acid (C20:4 n6 AA), and docosapentaenoic acid (C22:5n3 DPA) both fatty acids are important for your activity on human health. The missing of them make an incomplete data.
  • In meat quality chapter is not easy understand the effect of storage time on meat quality parameters, it is suggest to insert some number and refer it a specific table for the storage time.
  • Line 268: remove 1.
  • In fatty acid composition chapter is better, for explain the results, to follow the table trend, starting from SFA until the PUFA, hence it is suggested to rework it.
  • In table 8 form C18:3n3 to C22:6n3 check the correct marks.
  • Line 355: avoid to start the sentence wit “Next”.
  • In the "discussion" section the discussions should follow the trend of the results, starting from the “grow performance” section to the “amino acids”; so the writing is more harmonious.
  • References, check the correct format.

Reviewer 2 Report

Comments and Suggestions for Authors

The present manuscript investigates the effects of dietary n-3 polyunsaturated fatty acids (PUFA) and selenomethionine (SeMet) on meat quality and fatty acid composition in finishing pigs. This study addresses a significant topic in the fields of animal nutrition and food science, namely the enhancement of pork nutritional value.The experimental design is well-structured, incorporating multiple treatment groups and appropriate biochemical analyses. Nevertheless, there are notable weaknesses in clarity, statistical reporting and scientific interpretation that must be addressed prior to publication.

Major Concerns

The manuscript states that data were analysed using one-way and two-way ANOVA, yet specific details concerning post-hoc tests and assumptions (normality and homogeneity of variance) remain absent.
P-values are occasionally reported with excessive decimal places (e.g. P = 0.057). These should be rounded to two decimal places. 
Furthermore, the discussion would benefit from the inclusion of graphical representations, such as interaction plots, to enhance clarity.
The study does not provide a sufficient justification for the use of 3% supplementation levels of linseed and fish oil. The rationale behind selecting these levels warrants further elaboration, particularly whether they were based on prior literature or preliminary trials.The selection of 0.3 mg/kg SeMet lacks robust justification. Citations of studies that support this dosage would be beneficial.
Additionally, some claims made in the discussion are overly strong given the data presented. For instance, the assertion that SeMet supplementation "intensifies the nutritional and sensory qualities of pork" is not adequately substantiated by the findings.A more extensive comparison with existing studies should be incorporated into the discussion.Are the conclusions consistent with previous research? If discrepancies exist, what factors might account for them?The physiological mechanisms underlying SeMet's effects on fatty acid composition and antioxidant capacity are briefly mentioned but require more detailed elucidation.
The manuscript also lacks a clear explanation of how meat color, marbling, and pH measurements were standardized.The reported increase in L* value with SeMet supplementation is statistically significant but its biological relevance is unclear, and further discussion is needed on whether these changes are perceptible to consumers. 
Some of the tables are difficult to interpret due to excessive numerical values, and it is recommended that they be summarised or figures used where appropriate.Axes labels and legends in figures should be more informative.

Minor Concerns

The manuscript contains a number of typographical errors, including the use of incorrect abbreviations and inconsistent capitalisation. For instance, 'intense the nutritional qualities' should be 'enhance the nutritional qualities'.
Additionally, there is a lack of clarity in the use of abbreviations, such as 'n-3 PUFA' versus 'n-3PUFA'.
The Introduction could be improved by focusing more explicitly on the scientific gap and objectives.
While the health implications of n-3 PUFA are emphasised, the relevance to pig production and consumer acceptance should also be discussed.

Comments on the Quality of English Language

The manuscript contains a multitude of grammatical inaccuracies and sentence structures that compromise reader comprehension. A comprehensive language revision by a native English speaker or a professional editor is strongly advised.
Specific sections, notably the Introduction and the Results, exhibit redundancy or ambiguity in phrasing. To enhance conciseness and clarity, sentences should be restructured.

Round 2

Reviewer 1 Report

Comments and Suggestions for Authors

Dear Authors, have made a good job, but somethings need still to improve, to complete the article.

- You answered at my question on expression unit for AA and fatty acids as:

Thanks for reviewer’s suggestion. The peak amino acid was quantified by external standard method, the unit expression had been rewritten in Line182-190.  

But I am not able to check the lines, the pdf of reviewed version does not include the lines number, anyways I meant to add the unit expression in the “material and methods” chapter as:

“The 0.22 μm microporous filter membrane was filtered and analyzed with an automatic amino acid analyzer (Hitachi L-8900, Japan). The amino acids were express as % on total AA or  protein or mg/100g of meat or g/100 g of DM”

-Figures 2, 3, 4, 5: format the writing.

-In the note of table 6 you need to wrap the paragraph title.

-Table 7: I suggest using the same number of decimal places after comma for all data

- You answered at my question about the arachidonic acid (C20:4 n6 AA)

“I am sorry that arachidonic acid was not detected in our Experiment”

I think that you have confused, in the identify process, the C20:3n6 with C20:4n6. If you check the references is not possible that arachidonic acid is not detect in pig meat. I suggest to check the correct data.

Reviewer 2 Report

Comments and Suggestions for Authors

The manuscript was duly corrected.

Author Response

Dear Editor: Thank you very much for your comments and suggestions. I have learned a lot in the process of revising the manuscript. Thank you again and wish you all the best.